# The Isolation and Characterization of a Novel Group III-Classified Getah Virus from a Commercial Modified Live Vaccine against PRRSV

**DOI:** 10.3390/v15102090

**Published:** 2023-10-14

**Authors:** Xintao Gao, Jialei Li, Tong Wu, Jinping Dou, Wenrong Zhang, Hong Jia, Zhifang Zhang, Xingjian Liu, Yinü Li

**Affiliations:** 1Biotechnology Research Institute, Chinese Academy of Agricultural Sciences, Beijing 100081, China; xintao.gao@hotmail.com (X.G.); lijialei04@foxmail.com (J.L.); wutong01@caas.cn (T.W.); idoujinping@163.com (J.D.); bri-zhangzhifang@caas.cn (Z.Z.); 2College of Life Sciences, Capital Normal University, Beijing 100081, China; zhangwr111@163.com; 3Institute of Animal Sciences, Chinese Academy of Agricultural Sciences, Beijing 100081, China; jiahong80@126.com

**Keywords:** Getah virus, PRRSV vaccine, phylogenetic analysis, pathogenicity, mutation and glycosylation, E2

## Abstract

As an epizootic causative agent, the Getah virus (GETV) can cause moderate illness in horses, lethal disease in foxes, and reproductive disorders and fetal death in pigs. Due to the wide range of hosts and multiple routes of transmission, GETV has become a growing potential threat to the global livestock industry, and even to public health. More attention and research on GETV are urgently needed. In this study, we successfully isolated a novel GETV strain, named BJ0304, from a commercial live vaccine against porcine reproductive and respiratory syndrome virus (PRRSV) and determined its growth kinetics. Then, genetic and phylogenetic analyses were performed. The results revealed that BJ0304 was clustered into Group III, and it was most related to the GETV-V1 strain based on the complete genome sequence. Furthermore, the pathogenicity of the isolate was assessed and found to be a low virulent strain in mice relative to its closest homolog GETV-V1. Finally, mutation and glycosylation analysis showed that a unique mutation (171 T > I) at one amino acid of E2, which affected the glycosylation of E2, may be associated with viral pathogenicity. In summary, the general characteristic of a novel Group III-classified GETV-BJ0304 isolated from commercial live PRRSV vaccine was defined and then mutation/glycosylation-related potential virulence factor was discussed. This study highlights the complexity of GETV transmission routes in swine and the need for more surveillance on commercial animal vaccines, contributes to the understanding of genetic characterization of clinical isolates, provides possible virulence factors in favor of unveiling the viral pathogenesis, and eventually lays the foundation for the prevention and control of GETV.

## 1. Introduction

Getah virus (GETV) is a neglected mosquito-borne arbovirus belonging to the genus Alphavirus, family Togaviridae [1]. GETV has a single-stranded positive-sense RNA genome of ~11.7 kb with a 5′ terminal cap, two open reading frames (ORFs), a 26S junction region, and a 3′ poly(A) tail, with the two ORFs being responsible for encoding nonstructural (nsP1, nsP2, nsP3, and nsP4) and structural proteins (C, E3, E2, 6 K, and E1) [2,3]. For their role, in general, structural proteins mainly play a role in adsorption, assembly, and budding, while nonstructural proteins may exert the function of viral RNA replication. For example, the glycoproteins E1 and E2 form a spike in the viral envelope, which plays an important role in virus binding to cell surface receptors and membrane fusion [4].

GETV has a wide range of hosts, in which it can not only asymptomatically infect many vertebrate and arthropod species but also cause diseases in horse [5,6], pig [7], fox [8], and possibly even human [9,10] species. Since the first isolation of the GETV strain from Malaysia in 1955 [11], GETV has spread over a broad geographical area, including Europe, Asia, and Oceania [12]; in China, the virus was also first isolated from Culex mosquitoes collected in Hainan Province in 1964 [13]. Recently, the potential threats we face to GETV seem to be exacerbating, especially in China. Firstly, except for primary *Aedes* and *Culex* mosquitoes, viral mosquito vectors have expanded to many new mosquito species, including *Aedex vexans*, *Arigeres obturbans*, *Arigeres subalbatus*, and *Anopheles sinensis*, which are widely distributed in China, with large populations [14]. Secondly, since the beginning of the 21st century, sporadic cases or infection reports have been occurring more frequently in domestic animals such as pigs, horses, cattle, and foxes [15], among which the more susceptible pig species appear to be affected more adversely in China since the country is considered to have the largest pig industry in the world; notably, GETV contamination has been reported in a widely used commercial live PRRSV vaccine [12], indicating the complexity of GETV transmission routes in swine herd and the importance of surveillance on porcine vaccines. Therefore, more studies should be conducted on topics such as epidemiological analysis, genetic characteristics, and pathogenesis of the virus based on its wide host range, complex transmission routes, and potential contamination.

In the current study, a novel GETV strain, detected from a modified live vaccine (MLV) against PRRSV, was successfully isolated (BJ0304; GenBank: OM363683). Subsequently, the growth kinetics, whole-genome sequencing, and phylogenetic analysis were carried out to define its hallmark. Finally, the correlation between the mutation/glycosylation of nsP3/E2 and virulence was analyzed.

## 2. Materials and Methods

### 2.1. Virus Detection

Viral RNA in commercial modified live vaccines (MLVs) against PRRSV was extracted using a TIANamp Virus RNA Kit (TIANGEN, Beijing, China) and used for cDNA synthesis using HiScript^®^II 1st Strand cDNA Synthesis Kit (+gDNA wiper) (Vazyme, Nanjing, China). PCR was performed using Phanta Max Super-Fidelity DNA Polymerase (Vazyme, China) to detect GETV and PRRSV using the specific primers (nsp3-F and nsp3-R, N-F and N-R) designed with Primer Premier 6 (Appendix A).

### 2.2. Virus Isolation and Purification

The MLV diluted in PBS were filtered through 0.22 μm filters and then inoculated into swine testis (ST) cell monolayers in a six-well culture plate (Corning Life Sciences (Wujiang) Co., Ltd., Wujiang, China) for 1.5 h at 37 °C with 5% CO_2_. After washing, the cells were cultured in DMEM with 2% FBS and monitored daily. When obvious cytopathic effects (CPEs) were observed, cell cultures were frozen and thawed three times and centrifuged at 12,000 rpm for 10 min at 4 °C to collect the supernatant for further use.

Viral samples were serially diluted 10-fold with DMEM and then inoculated into ST cell monolayers for 1.5 h at 37 °C with 5% CO_2_. After that, the cells were overlaid using DMEM (2% FBS + 1% low melting agarose) for 1 h at room temperature to make the DMEM completely solidified. At 48 h after infection (hpi), the cells were stained with 0.01% neutral red solution for 1 h at 37 °C, and then the excess neutral red solution was discarded. The marked plaque together with DMEM was absorbed with a pipette, dissolved in serum-free DMEM, and frozen and thawed three times to fully release the virus before being inoculated for virus proliferation.

### 2.3. Antibody Production

The E2 nucleic acid was synthesized after obtaining the conserved E2 amino acid sequence (Appendix A) through alignment and optimization analysis, and its extracellular domain (ECD) was amplified via PCR using the primers E2 designed by SnapGene (Appendix A). The PCR products were digested using BamH I and EcoR I to construct the plasmid pET-28a-E2-ECD, which was then transformed into Escherichia coli BL21 (DE3) competent cells for expression with 1 mM IPTG for 6 h. The recombinant E2-ECD protein were purified via affinity chromatography. E2-ECD protein were mainly expressed in the form of inclusion bodies and purified using BeyoGold™ His-tag Purification Resin (P2233) (Beyotime, Shanghai, China) after 8 M urea denaturation, and then reconstituted and desalted via gradient dialysis. The purified protein was performed with endotoxin removal twice and confirmed using Western blot. Balb/c mice were injected with the mixture of purified recombinant E2-ECD and complete Freund’s adjuvant (1:1) to produce the anti-GETV-E2 polyclonal antibody. The anti-PRRSV-N polyclonal antibody was purchased from GeneTex, Irvine, California, USA. After inoculation, an ELISA assay to quantify anti-E2-ECD antibodies was performed. In brief, 100 µL diluted E2-ECD protein (5 μg/mL) was added to polyvinyl 96-well microtiter plates and incubated at 4 °C overnight. The wells were washed three times with PBST and incubated with 300 µL 5% skimmed milk for 3 h at 37 °C. After that, 100 µL serum serial dilutions were added to the wells and incubated for 1.5 h at 37 °C. Following washing, 100 µL HRP-conjugated goat anti-mouse IgG (1:6000) (Bioss, Beijing, China) was added to the wells and incubated at 37 °C for 1 h. After four washes, 100 µL OPD solution was added to every well and incubated for 30 min at 25 °C. Finally, 50 µL of 2 M H_2_SO_4_ was added to stop the reaction. The 450 nm optical density of the plate was read with a microplate reader (Thermo Fisher Scientific Multiskan MK3, Shanghai, China). Values over the cut-off background level (mean value of saline group multiplied by 2.1) were considered positive, and the reciprocal end dilutions of sera represented the titers.

### 2.4. Indirect Immunofluorescence Assay (IFA)

ST cell monolayers were inoculated with the novel GETV strain at a multiplicity of infection (MOI) of 0.1 for 1.5 h and then cultured with DMEM with 2% FBS for 24 h before analyzing the GETV infection using IFA as described by Li et al. [16].

### 2.5. Virus Growth Kinetics

ST cell monolayers were infected by the novel isolated GETV at MOI of 0.05, 0.1, and 0.2. After 1.5 h, the inoculated cells were incubated in the media (DMEM+2% FBS) at 37 °C with 5% CO_2_. At different time points after infection, the cells were harvested for detecting the copies of GETV using nsp3 primers (Appendix A) via RT-qPCR, and the supernatant was collected for virus titers assay according to Reed and Muench method [17]. The inoculation of serially diluted viral suspension, following a logarithmic gradient, was administered onto a 96-well microtiter plate containing ST cells. Each serial dilution was performed with 8 replicates. Subsequent to inoculation, the daily scrutiny of cellular dynamics was conducted utilizing optical microscopy, whereby meticulous accounts of pathological manifestations were diligently transcribed. Finally, the 50% tissue culture infectious dose (TCID_50_) was determined using the Reed and Muench method.

### 2.6. Determination of GETV Complete Genome

Viral RNA was extracted from the supernatant of virus-infected cell cultures and reversely transcribed into cDNA. Eleven specific primers targeting conserved regions were designed using SnapGene (Appendix A) to amplify and sequence the complete genome of the virus. Meanwhile, the 5’UTR and 3’UTR of the viral genomic RNA were amplified through HiScript-TS 5’/3’ RACE Kit (Vazyme, Nanjing, China), and the amplified products were cloned into pCE2 TA/Blunt-Zero vector (Vazyme, China) before being sequenced using M13 primers. These contig sequences were assembled using DNAMAN 6.0.3.99 software.

### 2.7. Virus Infection

Twelve six-week-old male specific-pathogen-free mice with a body weight of 20–23 g were randomly divided into two groups (*n* = 5) and intramuscularly injected with 100 μL (100 × TCID_50_/0.1 mL, 50% tissue culture infectious dose) GETV-BJ0304 strain, or control DEME. At 0.5–14 days post-infection (dpi), clinical symptom monitoring and blood collection were performed in all mice. Organs of mice were harvested at 7 and 14 dpi for further analysis. Viral RNAs were extracted with Trizol reagent (Invitrogen, Shanghai, China) and determined via PCR performed using Phanta Max Super-Fidelity DNA Polymerase before cDNA synthesis using HiScript^®^II 1st Strand cDNA Synthesis Kit (+gDNA wiper). Mice were obtained from Beijing Vital River Laboratory Animal Technology Co., Ltd. (Beijing, China).

### 2.8. Sequence Alignments and Phylogenetic Analysis

Using Simplot version 3.5.1, Mega 7.0, MegAlign 7.1.0.44, BioEdit 7.0.9.0, and other software, the homology and phylogenetic analyses were performed in the novel GETV strain and 20 representative GETV strains obtained from GenBank (Appendix A) based on the complete genome and E2 nucleotide sequences. A maximum likelihood phylogenetic tree based on the E2 gene was constructed using the Tamura–Nei model [18], while a maximum likelihood phylogenetic tree based on the complete genome was constructed using the general time reversible model [19]. Evolutionary analyses were conducted in MEGA7 [20].

### 2.9. Glycosylation Analysis in E2

*N*-glycosylation and O-glycosylation sites in the E2 amino acid sequence were predicted using the NetNGlyc-1.0 server (https://services.healthtech.dtu.dk/service.php?NetNGlyc-1.0, accessed on 18 April 2022) and NetOGlyc-4.0 server (https://services.healthtech.dtu.dk/service.php?NetOGlyc-4.0 accessed on 18 April 2022), respectively.

### 2.10. Statistics Analysis

Data were analyzed using GraphPad Prism 8.4.3. Comparisons were performed using two-way ANOVA. Differences were considered significant if the *p* value was <0.05, and *p* values are indicated as follows: ns > 0.05; * *p* < 0.05; ** *p* < 0.01; *** *p* < 0.001; **** *p* < 0.0001.

## 3. Results

### 3.1. Anti-GETV-E2 Polyclonal Antibody Production

After determining that E2-ECD was successfully expressed in the form of inclusion bodies (Figure 1A), the protein was purified and confirmed via Western blot analysis (Figure 1B) and liquid chromatography–tandem mass spectrometry (LC-MS/MS) (Appendix A). Afterward, the purified E2-ECD mixed with complete Freund’s adjuvant at a ratio of 1:1 was inoculated into per mouse with 20 µg every two weeks. Serum was collected two weeks after inoculation, and then the titer of E2-ECD-specific IgG antibody in mouse serum was measured using enzyme-linked immunosorbent assay (ELISA). The results showed that the antibody titer significantly increased with the prolongation of time and reached more than 10^5.3^ 42 days after the first immunization (Figure 1C). Furthermore, the indirect immunofluorescence assay (IFA) method was successfully established by using the prepared anti-GETV-E2 polyclonal antibody as the primary antibody (Figure 1D).

### 3.2. Isolation and Identification of Isolated GETV Strain

Of the several PRRSV commercial modified live vaccines (MLVs) from the same batch, one sample was found to be GETV-positive via specific RT-PCR. Subsequently, a new isolate, named BJ0304, was obtained in ST cells after serial passages and plaque purification (Figure 2A,B), and CPEs, including cell aggregation and shrinking, followed by detachment, were observed in the infected ST cell monolayers (Figure 2C).

To determine viral growth kinetics, an 80% confluent cellular monolayer of ST cells was infected (MOI = 0.05/0.1/0.2) to detect the copy number and viral titer at indicated time points. The results showed that when MOI = 0.1, the RNA replication and titer of BJ0304 exhibited a relatively consistent trend, that is, a rapid increase from 6 hpi and a peak at 36 hpi (Figure 2D,E). Taken together, these results revealed that a GETV strain was successfully isolated from a commercial PRRSV MLV using ST cells, and a GETV infection model in ST cells was developed.

### 3.3. Genetic and Phylogenetic Analyses of Isolated GETV

The genome sequences of the novel isolated GETV were amplified, sequenced, and assembled by using primers targeting 11 overlapping cDNA fragments and 3′/5′-RACE. The full-length genome sequence, which was submitted to GenBank (accession number: OM363683) and named GETV-BJ0304, was 11,689 nt in size excluding the poly(A) tail, composed of 5′UTR (78 nt, 1–8), 3′UTR (401 nt, 11,289–11,689), 26S junction region (44 nt, 7483–7526), and two ORFs encoding nonstructural polyprotein (7404 nt, 79–7482) and structural polyprotein (3762 nt, 7527–11,288); the nonstructural polyprotein was further cleaved into four proteins in the order of nsP1 (1602 nt, 79–1680), nsP2 (2394 nt, 1681–4074), nsP3 (1572 nt, 4075–5646), and nsP4 (1833 nt, 5647–7479), while the structural polyprotein was cleaved into five proteins, namely C (804 nt, 7527–8330), E3 (192 nt, 8331–8522), E2 (1266 nt, 8523–9788), 6K (183 nt, 9789–9971), and E1 (1317 nt, 9972–11,288), one by one (Figure 3A).

The sequence alignment among strains revealed that BJ0304 shared the highest identities with the Republic of Korea 2004 strain at the nucleotide level of the complete genome (99.4%) and at the amino acid level of E2 (99.8%), while the corresponding homology with our intentional reference strain (GETV-V1), first isolated from a commercial PRRSV MLV in China, were 99.1% and 99.3%, respectively (Appendix A).

Phylogenetic analysis of the E2 showed that BJ0304 was clustered into Group III and was most similar to the Republic of Korea strain (Figure 3C), while the phylogenetic analysis of the complete genome revealed that the strain was most related to the GETV-V1 strain (Figure 3B).

### 3.4. Pathogenicity of the GETV-BJ0304 Strain

Mice were inoculated intramuscularly with 100 μL (100 × TCID_50_/0.1 mL) of the GETV-BJ0304 strain, and clinical signs were observed daily. No clinical signs (i.e., weight loss, fever, paralysis in pelvic limbs, or rash) were observed when mice were inoculated with GETV at 0.5–14 dpi. At 7 and 14 dpi, there were no obvious histopathological changes in the tissue samples (i.e., testis, brains, lungs, spleens, kidneys, livers, and hearts) of the mice. To further investigate the epidemic GETV infection, we performed RT-PCR by using RNA from all mice samples. Only testis and kidney samples from GETV-inoculated mice were positive at 7 and 14 dpi (Appendix A).

### 3.5. Recombination and Mutation Analysis of GETV

Overall, no recombination event was found within GETV strains using Simplot analysis. As shown in Figure 4, GETV-BJ0304 exhibited over 90% nucleotide sequence identity to other strains of Group I-IV from a genome-wide sequence perspective, only showing the divergence to some extent from the strains of Group I at the nsP3 position.

Compared with the other 20 GETV strains, GETV strain BJ0304 possessed 44 specific nucleotide substitutions, leading to 10 amino acid substitutions, including 7 substitutions in nonstructural proteins and 3 substitutions in structural proteins. Among the nonstructural proteins, the largest number of amino acid substitutions was found in the nsP3 protein, which had three substitutions, two of which were within the hypervariable region located on the C-terminus of the protein. Among the structural proteins, the amino acid substitutions were relatively limited, with one in the C protein and two in the E2 protein (Table 1).

### 3.6. Glycosylation Sites at E2

The glycosylation of viral proteins plays an important role in the viral life cycle such as immune evasion, virulence, and pathogenicity [21,22]. To investigate whether the modification occurs at E2 of GETV-BJ0304, the *N*-glycosylation and *O*-glycosylation sites of E2 were predicted using the NetNGlyc-1.0 and NetOGlyc-4.0 servers, respectively. As shown in Table 2, two *N*-glycosylation sites (N200 and N262) and three *O*-glycosylation sites (T142, T159, and T160) at E2 were predicted. Compared with GETV-V1, three *O*-glycosylation sites (T154, T155, and T163) were lacking in BJ0304, which may be related to the difference of the amino acid residue at E2 between BJ0304 (171I) and GETV-V1 (171T) (Table 2).

## 4. Discussion

GETV, an arbovirus, has been detected in several mosquito species (i.e., *Aedex vexans*, *Armigeres obturbans*, *Armigeres subalbatus*, and *Anopheles sinensis*), almost all of which are widely distributed in China, with large populations [14]. Meanwhile, GETV is an unequivocal causative agent of several diseases occurring in horse, fox, and pig species with different degrees of clinical symptoms varying from body rash, leg edema, fever, and reproductive disorders to fetal death [6,7,8].

Since the first epizootic GETV infection was reported in 1978 [23], outbreaks of horse and pig infections usually occurred in Japan and India in the following two decades [24]. In China, it has spread rapidly since the 21st century and has been distributed in 24 provinces to date [25], concomitant with the accelerated outbreaks in horse, pig, and fox species. Among these, the situation appears to be most severe in pigs, as this animal itself seems to play a pivotal role in virus transmission given the fact that the highly pathogenic swine GETV can cause this lethal disease in foxes [8]. In 2020, Zhou et al. first reported the isolation and identification of a porcine strain GETV-V1 from a commercial live vaccine against PRRSV used in China [12], and this GETV contamination was also observed in this study (Figure 2). These two independent cases indicated that contamination is not an isolated incident and could be widespread to some extent in China. This putative but rigorous situation increases the transmission routes of the virus and the complexity of epidemiological investigation, thereby making the prevention and control of GETV more difficult. Furthermore, although the Simplot analysis in this study implied that the probability of recombination between GETV Group III and other groups was low (Figure 4), the contamination source still increases the possibility of the recombination and mutation within GETVs or other alphaviruses due to the role of pigs in the viral reservoir and because China has the largest pig industry in the world. Hence, the need for more surveillance of commercial animal vaccines and more monitoring of the genetic characterization of clinical isolates should be emphasized.

The nsP3 protein is considered to be associated with RNA synthesis and may be an important virulence factor [26]. E2, a highly conserved transmembrane glycoprotein of alphavirus, has also been reported to be of importance for viral infection, budding, assembly, and pathogenicity [27,28]. For example, scientists reported that viral replication can be impeded through the inhibition of interactions between nsP3 and host proteins, including stress granule-related proteins, dead box proteins, heat shock proteins, and kinases [29,30,31]. Also, multiple studies have shown that creating mutations or deletions within the nsP3 hinders the viral ability to effectively replicate in the host [32,33,34,35]. E2 is responsible for mediating receptor binding during viral infection [27] and has been reported to facilitate virus particle assembly and disassembly by interacting with capsid protein [27]. In this study, 10 specific amino acid substitutions were observed in nsP1, nsP2, nsP3, nsP4, C, and E2 proteins by comparing the full-length sequences of BJ0304 with other GETV strains (Table 1). Of those, two amino acid mutations in E2 and three in nsP3 triggered our interest based on the dominant viral virulence-related functions E2 and nsP3 exerted. Since the GETV-BJ0304 we isolated showed low virulence and pathogenicity in mice (Appendix A), the amino acid sequence of nsP3 and E2 between highly pathogenic swine GETV isolates from GenBank and GETV-BJ0304 was further analyzed. The specific amino acid substitutions were observed at S/G (30) and E/D (444) of nsP3 as well as E/K (4) and T/I (171) of E2 (Appendix A), which are almost consistent with the results of the alignment presented in Table 1, implying that the mutations (4/5) occurring in GETV-BJ0304 are specific and potentially providing a possible explanation for the low virulence of the isolate in mice. On the other hand, previous studies reported that the glycosylation of E2 was a feature of alphaviruses related to virulence and cellular tropism [36,37]. Wang et al. also found that some glycan moieties at E2 of GETV were surface-exposed, suggesting that E2 glycosylation may be beneficial for viral immune evasion by preventing host detection [38]. GETV-V1, the most closely related strain to GETV-BJ0304 according to the phylogenetic analysis of the complete genome, was also isolated from a commercial modified live vaccine and could cause serious lesions in mice [38]. Since GETV-BJ0304 and GETV-V1 have the same source and quite high homology but different degrees of virulence, glycosylation sites at the E2 of the two strains were also predicted in this study. The results showed that glycosylation occurred at five sites in GETV-BJ0304, namely T142/T159/T160 and N200/N262, lacking three O-glycosylation sites compared with GETV-V1 (Table 2). Interestingly, these three O-glycosylation sites were supplemented after changing one amino acid residue of BJ0304-E2 from I171 to T171 in the NetOGlyc-4.0 server, and this intended mutation was also observed in alignment analysis between BJ0304 (I171) and GETV-V1 (T171) (Table 1), leading us to the speculation that this unusual mutation (171 T > I) affects the glycosylation of E2 and thus may affect viral pathogenicity. Therefore, we hypothesized that the position at which mutation and glycosylation occur may be one of the key factors/sites for the functions nsp3 and/or E2 serving, which deserves further investigation.

One drawback of this study is that the pig infection experiment was not performed temporarily due to the limited pig breeding conditions and difficulty in the acquisition of PRRSV-free animals. However, this assay is important for understanding the real pathogenicity of the novel isolates and comparing the clinical symptoms with the side effects or immune stress response that are often induced by the live PRRSV vaccine, like fever, anepithymia, and miscarriage. Hence, this issue needs to be addressed in the future.

## 5. Conclusions

A novel Group III-classified GETV-BJ0304 from commercial live PRRSV vaccine was successfully isolated, and its general characteristic was defined, highlighting the need for more surveillance on commercial animal vaccines, contributing to a better understanding of the genetic characterization of clinical isolates, and providing possible virulence factors to unveil the pathogenesis of GETV.

## Figures and Tables

**Figure 1 viruses-15-02090-f001:**
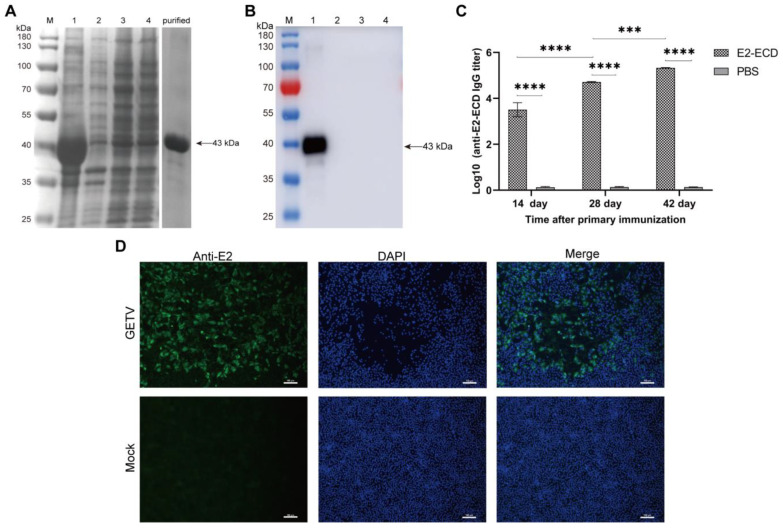
Expression of E2-ECD protein and preparation of its polyclonal antibody: (**A**) SDS-PAGE analysis of recombinant pET-28a-E2-ECD expressed in *E. coli* BL21 (DE3). Lane M: protein marker; Lane 1, precipitate of pET-28a-E2-ECD samples; Lane 2, precipitate of pET-28a samples; Lane 3, supernatant of pET-28a-E2-ECD samples; Lane 4, supernatant of pET-28a samples; Lane 5, recombinant E2-ECD proteins purified by Ni-NTA. (**B**) Western blot analysis of recombinant pET-28a-E2-ECD expressed in *E. coli* BL21 (DE3). Lane M: protein marker; Lane 1, precipitate of pET-28a-E2-ECD samples; Lane 2, precipitate of pET-28a samples; Lane 3, supernatant of pET-28a-E2-ECD samples; Lane 4, supernatant of pET-28a samples. (**C**) Determination of serum antibody titer in mice immunized with the mixture of purified recombinant E2-ECD and complete Freund’s adjuvant (1:1). The results are presented as the mean ± SD (*n* = 3); *p* values are indicated as follows: ns > 0.05; *** *p* < 0.001; **** *p* < 0.0001. (**D**) Determination of the reactivity of polyclonal antibodies against E2-ECD protein to GETV by IFA. Scale bars = 100 µm.

**Figure 2 viruses-15-02090-f002:**
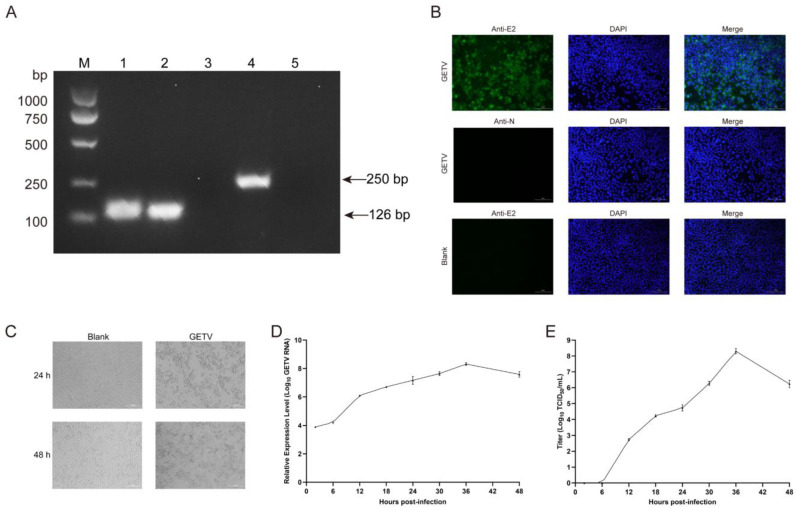
Isolation and identification of Getah virus (GETV) BJ0304 strain grown in swine testis (ST) cells: (**A**,**B**) RT-PCR and immunofluorescence assay (IFA) were performed to verify the purity of GETV sample after purification. Lane M, DNA marker; Lane 1, viral sample with specific primers of GETV nsp3; Lane 2, positive control of GETV with specific primers of GETV nsp3; Lane 3, viral sample with specific primers of PRRSV N; Lane 4, positive control of PRRSV with specific primers of PRRSV N; Lane 5, negative control. The infected ST cells were stained using an anti-GETV-E2 polyclonal antibody or anti-PRRSV-N polyclonal antibody and CF^®^488A goat anti-mouse IgG (H+L) (green) or CF^®^488A goat anti-rabbit IgG (H+L) (green) before being stained with DAPI (blue). Images were taken using a 20× objective. (**C**) Cytopathic effects (CPEs) in ST cells infected with the GETV BJ0304 strain at 24 hpi and 48 hpi. (**D**,**E**) The RNA replication and titers of BJ0304 at different time points when ST cells were infected with MOI = 0.1. The RNA replication was detected by examining the transcriptional level of nsp3 using RT-qPCR, and the titer of BJ0304 was determined using TCID_50_ assay. The results are presented as the mean ± SD (*n* = 3).

**Figure 3 viruses-15-02090-f003:**
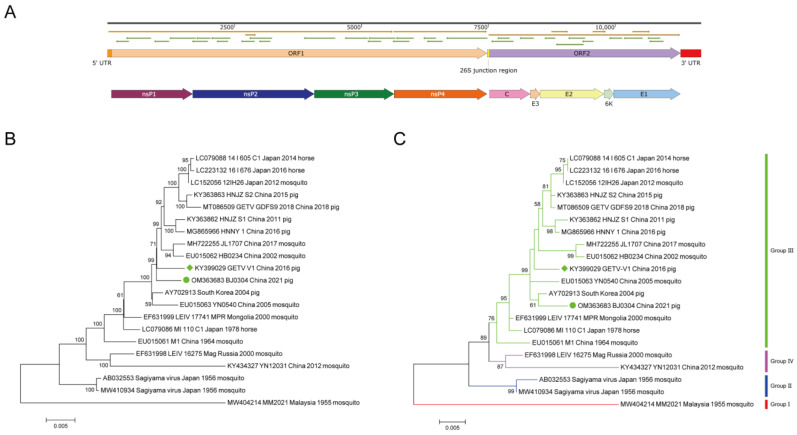
Genomic characteristics and phylogenetic analysis of GETV: (**A**) Genomic structure of GETV BJ0304 strain. The different color region stands for indicated ORF, UTR, or gene. (**B**) Phylogenetic analysis among BJ0304 and other GETV strains based on the complete genome sequence using the maximum likelihood method based on the general time reversible model (left) and (**C**) E2 nucleotide sequences using the maximum likelihood method based on the Tamura–Nei model (right).

**Figure 4 viruses-15-02090-f004:**
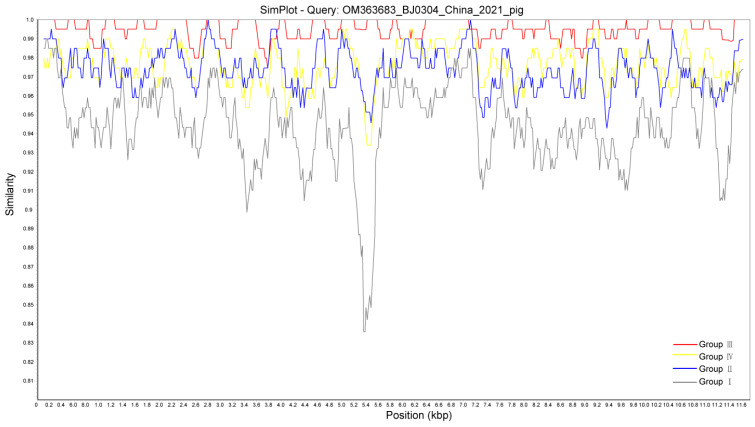
Simplot analysis of GETV strains. The similarity of nucleotide sequences of BJ0304 with other GETV strains was determined using Simplot version 3.5.1 with a sliding window of 200 bp and a step size of 20 bp.

**Table 1 viruses-15-02090-t001:** Unique nucleotide mutations in BJ0304 strain genome compared with other strains.

Gene	Position	Mutation (nt)	Mutation (aa)	Gene	Position	Mutation (nt)	Mutation (aa)
nsP1	398	T/C	L/P (107)	C	7532	T/C	I/L (35)
	720	A/T	-		7629	A/C	-
	954	C/T	-		7907	C/T	-
	957	G/A	-		8069	T/C	-
nsP2	1854	C/T	-		8243	C/T	-
	2544	T/C	-	E2	8532	G/A	E/K (4)
	2583	T/C	-		8687	C/T	-
	2676	G/A	-		8927	A/G	-
	3183	C/T	-		8930	A/T	-
	3698	T/C	M/T (673)		9034	C/T	T/I (171)
	3714	C/T	-		9338	A/G	-
	3727	G/A	V/I (683)		9434	T/C	-
	3870	C/T	-		9566	T/C	-
nsP3	4162	A/G	S/G (30)	6K	9790	G/A	-
	4386	C/T	-		9923	G/A	-
	4413	C/A	-	E1	10,337	C/T	-
	4830	T/C	-		10,859	C/T	-
	4884	G/T	-		11,159	C/T	-
	5303	C/T	A/V (410)	3’UTR	11,345	A/G	-
	5406	A/T	E/D (444)		11,347	A/C	-
nsP4	5694	C/T	-				
	6015	C/T	-				
	7324	A/G	I/V (560)				
	7347	C/T	-				

Notes: T/C represents nucleotide mutation From T to C, L/P (107) represents the 107th amino acid of the protein varied from L to P; - indicates no mutation of amino acid.

**Table 2 viruses-15-02090-t002:** Glycosylation sites in E2.

Glycosylation Type	Glycosylation Site in BJ0304	Glycosylation Site in GETV-V1
*O*-Glycosylation	T142/T159/T160	T142/T154/T155/T159/T160/T163
*N*-Glycosylation	N200/N262	N200/N262

## Data Availability

Data are available upon request.

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
