# Peer review of "The Isolation and Characterization of a Novel Group III-Classified Getah Virus from a Commercial Modified Live Vaccine against PRRSV"

_viruses, 2023, doi:10.3390/v15102090_

Round 1
Reviewer 1 Report
Manuscript viruses-2651252 by Gao et al. describes isolation and characterization of a novel GETV strain BJ0304 from a commercial live vaccine against PRRSV. The manuscript characterized the BJ0304 strain in multiple perspective including growth kinetics, whole genome sequencing, phylogenetic analysis, mutation analysis compared to other GETV strains, pathogenicity analysis and analysis of N- and O- glycosylation analysis.
This paper includes fundamental analysis of newly discovered GETA strain but I have some comments below :
Introduction: it is recommended that authors add some brief descriptions about the role or function of virally coded nsp and sp proteins for better understanding viral biology. It is also recommended that authors add some inputs of the association between commercial PRRSV and GETV, what is the initial intention to isolate GETV from live vaccine and why it's important. Are there any other viruses could possibly isolated from live PRRSV?
Section 2.3: Define E2-ECD before use.
Lines 111-112: paraphrase the sentence "A complement of eight replicate wells accompanies each dilution iteration". Do you mean by performing each serial dilution with 8 replicates?
Lines 163-165: it is recommended that authors to include experimental procedures of ELISA assay to quantify anti-E2-ECD antibodies in Materials and Methods.
Figure 1 Caption: it is recommended that authors to include detailed experimental procedures of how to purify E2-ECD protein in Materials and Methods. it's very confusing to read Lanes 1-4 in gel images without understanding the details of how E2-ECD was purified.
Lines 251-254: it is recommended that authors to annotate the small decrease in the partial nsP3 of Group I. What does that mean, do you mean by there is no template switching across strains in different groups of GETV?
Table 1: Why BJ0304 strain was compared to other 20 GETV strains not the parent strain of GETV. And which amino acid was mutated compared to which of the 20 GETV strains.
Line 323: I could'd find Table 3
The English is understandable to me but certainly can be improved e.g. in multiple places, at lines 146, 150, 156, "by using" was used, which can simply use "by".
Author Response
1. Summary |
|
|
Thank you very much for taking the time to review this manuscript. Please find the detailed responses below and the corresponding revisions/corrections highlighted in the re-submitted files.
|
||
2. Questions for General Evaluation |
Reviewer’s Evaluation |
Response and Revisions |
Does the introduction provide sufficient background and include all relevant references? |
Must be improved |
Thank you very much for your comment. According to your advice, we have added more content in Introduction. |
Are all the cited references relevant to the research? |
Yes |
Thank you very much for your comment. |
Is the research design appropriate? |
Yes |
Thank you very much for your comment. |
Are the methods adequately described? |
Can be improved |
Thank you very much for your comment. According to your advice, we have added more detailed experimental procedures in Materials and Methods. |
Are the results clearly presented? |
Can be improved |
Thank you very much for your comment. According to your advice, some Results were rewrote to improve the readability. |
Are the conclusions supported by the results? |
Yes |
Thank you very much for your comment. |
3. Point-by-point response to Comments and Suggestions for Authors |
||
Comments 1: Introduction: it is recommended that authors add some brief descriptions about the role or function of virally coded nsp and sp proteins for better understanding viral biology. It is also recommended that authors add some inputs of the association between commercial PRRSV and GETV, what is the initial intention to isolate GETV from live vaccine and why it's important. Are there any other viruses could possibly be isolated from live PRRSV?
|
||
Response 1: We appreciate the comments and suggestions. According to your advice, we have added the brief content about the function of viral structural and non-structural proteins at the end of the first paragraph of the Introduction in revised manuscript (Line 43-47 in the revised manuscript, previously Line 42). In 2020, Zhou et al. firstly isolated a GETV strain from a commercial live PRRSV vaccine which was widely used in China (Reference 12). Based on his findings, we also carried out the GETV surveillance of PRRSV vaccines and observed the same contamination in our study. These two independent cases indicated that the contamination is not an isolated incident and could be widespread to some extent in China. So far, no GETV contamination case was reported in other commercial live vaccines of swine or other animals except for PRRSV vaccine; and, we did not perform the surveillance on other animal-used commercial live vaccines neither. Considering the similar clinical symptoms, GETV contamination in commercial modified live vaccine against PRRSV seems to be more easily overlooked. However, other vaccines may also be contaminated with the virus, as the contamination was likely to come from trypsin and fetal bovine serum used for cell passage, which was rationally speculated by Zhou in reference 12. To date, GETV has not yet attracted enough attention from the relevant including the manufacturer, regulator and user. To prevent this potential transmission route, we think regulators should add GETV in the list of exogenous agents testing for veterinary vaccine manufacture, thereby facilitating routine GETV surveillance on animal-used commercial live vaccines. |
||
Comments 2: Section 2.3: Define E2-ECD before use. |
||
Response 2: Thank you very much for your comment. The abbreviation ECD was defined at line 98 in revised manuscript (previous line 93). |
||
Comments 3: Lines 111-112: paraphrase the sentence "A complement of eight replicate wells accompanies each dilution iteration". Do you mean by performing each serial dilution with 8 replicates? |
||
Response 3: Thank you very much for your comment. We are sorry for making you confused and have revised the sentence ‘A complement of eight replicate wells accompanies each dilution iteration’ to ‘Each serial dilution was performed with 8 replicates (Line 132-133 in the revised manuscript, previously Line 111-112). |
||
Comments 4: Lines 163-165: it is recommended that authors to include experimental procedures of ELISA assay to quantify anti-E2-ECD antibodies |
||
Response 4: Thank you very much for your comment. According to your advice, we have added experimental procedures of ELISA assay in 2.3 of Materials and Methods in revised manuscript (Line 109-120 in revised manuscript, previously Line 99). |
||
Comments 5: Figure 1 Caption: it is recommended that authors to include detailed experimental procedures of how to purify E2-ECD protein in Materials and Methods. it's very confusing to read Lanes 1-4 in gel images without understanding the details of how E2-ECD was purified. |
||
Response 5: Thank you very much for your comment. We have added detailed purification procedure after the sentence ‘The recombinant E2-ECD were purified by affinity chromatography’ in 2.3 of Materials and Methods in revised manuscript (Line 102-106 in revised manuscript, previously Line 96). Also, the sentences (Line 176-178 in revised manuscript, previously Line 155-161) in 3.1 of Results were rewrote to improve the readability. |
||
Comments 6: Lines 251-254: it is recommended that authors to annotate the small decrease in the partial nsP3 of Group I. What does that mean, do you mean by there is no template switching across strains in different groups of GETV? |
||
Response 6: Thank you very much for your comment. We are sorry for making you confused and have revised the whole paragraph to ‘Overall, no recombination event was found within GETV strains by Simplot analysis. As shown in Figure 4, GETV-BJ0304 exhibited over 90% nucleotide sequence identity to others strains of Group I-IV from a genome-wide sequence perspective, only showed the divergence to some extent from the strains of Group I at the nsP3 position’ (Line 268-271 in revised manuscript, previously Line 251-254). |
||
Comments 7: Table 1: Why BJ0304 strain was compared to other 20 GETV strains not the parent strain of GETV. And which amino acid was mutated compared to which of the 20 GETV strains. affinity chromatography |
||
Response 7: Thank you very much for your comments. We are sorry for making you confused. As we didn’t know the parent strain of GETV, we sought to found the unique nucleotide mutations in BJ0304 strain genome compared with other 20 strains which were selected from NCBI according their source such as host, country, and collection date. The information of amino acid mutations was listed in the column 4 and 8 of Table 1. For example, L/P (107) represents the 107th amino acid of the protein was leucine (L) in most other 20 GETV strains but proline (P) in BJ0304 strain. |
||
Comments 8: Line 323: I couldn't find Table 3 |
||
Response 8: Thank you very much for your comment. We are sorry for the mistake, and have revised the ‘Table 3’ to ‘Table S7’ in revised manuscript (Line 340 in revised manuscript, previously Line 323). |
||
|
||
4. Response to Comments on the Quality of English Language |
||
Point 1: The English is understandable to me but certainly can be improved e.g. in multiple places, at lines 146, 150, 156, "by using" was used, which can simply use "by". |
||
Response 1: Thank you very much for your comment. According to your comment, we have deleted ‘using’ at lines 167, 171, 177, 182, 268 in revised manuscript (previous Lines 146, 150, 156, 164, 251). Meanwhile, subscript characters in CO2 and TCID50 were corrected in revised manuscript. |
Reviewer 2 Report
This paper describes the characteristics of a new group-III Getah virus that was isolated from a commercial modified-live PRRSV vaccine in China. The virus was isolated and plaque purified on swine testis cells and this purified isolate was used as the new strain of virus. How did you separate the Getah isolate from the modified PRRS virus? Did you check your isolate for PRRSV contamination to rule this out?
Essentially, the authors describe the following characteristics of the Getah virus: cytopathic effect in cell culture; complete genome sequence; N and O glycosylation sites; immunogenicity of the E2 protein construct; and lack of pathogenicity in mice.
The study was simply designed to provide evidence that this Getah virus is a new strain or isolate. This was sufficiently proven by the methods used by the authors. Overall this is a good paper that provides additional evidence of the importance of Getah virus infections in China. The paper would have been strengthen by including experiments on the pathogenicity of this Getah isoalate in pigs. Unfortunately, this was not done in this study, but the authors discuss this as a weakness of their study in lines 342-351 of the paper. It might be helpful to provide some additional description of Getah virus infections in pigs in the Introduction of the paper.
Questions and concerns:
1. Authors can you provide information or data on how the modified-live PRRSV strain was contaminated with the Getah virus? What type of cells were used to propagate the vaccine virus?
2. In your supplementary Table on observed clinical signs and histopathology in mice, you list a description of lesions on some organs. Were these lesions or just a description of what you expected and how to you dismiss these microscopic lesions as not related to the Getah virus?
3. Section 2.5, authors you should reword the last sentence that the TCID 50 was meticulously computed. No doubt you were meticulous but it suffice to say that the TCID 50 was determined by the method of ??????
4. Section 3.1, line 166, "..... reached more that 105.3? Authors should this be 105.3?
5. In your phylogenetic diagrams in Fig 3, did not see your isolate BJ0304 in the list?
6. Is Getah virus normally pathogenic in mice or does it mostly result in asymptomatic infections? Your speculation that the E3 and nsp3 may be involved in pathogenicity is interesting and you do provide some preliminary evidence for this. However, there may be other reasons for why your isolate is not pathogenic in mice, can you suggest any other reasons?
7. Finally in your title, should it not be ".... from a commercial modified-live virus vaccine."
Very good overall, some minor editing would improve the communication in certain areas.
Author Response
1. Summary |
|
|
Thank you very much for taking the time to review this manuscript. Please find the detailed responses below and the corresponding revisions/corrections highlighted in the re-submitted files.
|
||
2. Questions for General Evaluation |
Reviewer’s Evaluation |
Response and Revisions |
Does the introduction provide sufficient background and include all relevant references? |
Yes |
Thank you very much for your comment. |
Are all the cited references relevant to the research? |
Yes |
Thank you very much for your comment. |
Is the research design appropriate? |
Can be improved |
Thank you very much for your comment and suggestion. We have revised some content in revised manuscript. |
Are the methods adequately described? |
Yes |
Thank you very much for your comment. |
Are the results clearly presented? |
Yes |
Thank you very much for your comment. |
Are the conclusions supported by the results? |
Yes |
Thank you very much for your comment. |
3. Point-by-point response to Comments and Suggestions for Authors |
||
Comments 1: Authors can you provide information or data on how the modified-live PRRSV strain was contaminated with the Getah virus? What type of cells were used to propagate the vaccine virus? |
||
Response 1: Thank you very much for your comments. To be honest, we are not sure how Getah virus (GETV) contamination happened. However, we tend to agree with the speculation that trypsin and fetal bovine serum for cell passage may cause the contamination, which proposed by Zhou (Reference 12). Once the contamination occurred, even with a high virulent strain to swine, it is not easy to be found because the clinical symptoms caused by GETV may be considered a side effect of the PRRSV vaccine, and also the virus has not yet attracted enough attention from the relevant including the manufacturer, regulator and user since GETV is not included in the list of adventitious agents testing for veterinary vaccine manufacture. Marc-145 cells were used to propagate Porcine reproductive and respiratory syndrome virus (PRRSV). |
||
Comments 2: In your supplementary Table on observed clinical signs and histopathology in mice, you list a description of lesions on some organs. Were these lesions or just a description of what you expected and how to you dismiss these microscopic lesions as not related to the Getah virus? |
||
Response 2: Thank you very much for your comment. We are sorry for making you confused. According to previous publications, high-virulence GETV strain infection typically can cause clinical symptoms and pathological lesions of certain organs as listed in table S7. However, no typical pronounced clinical symptoms were observed in this study, and also, pathological lesions did not occur in histopathology analysis as we expected. |
||
Comments 3: Section 2.5, authors you should reword the last sentence that the TCID 50 was meticulously computed. No doubt you were meticulous but it suffice to say that the TCID 50 was determined by the method of ?????? |
||
Response 3: Thank you very much for your comment and suggestion. We have revised the sentence ‘Subsequent to this, quantification of the 50% Tissue Culture Infectious Dose (TCID50) is meticulously computed’ to ‘Finally, the 50% tissue culture infectious dose (TCID50) was determined by Reed and Muench method’ (Line 135-136 in revised manuscript, previously Line 115). |
||
Comments 4: Section 3.1, line 166, "..... reached more that 105.3? Authors should this be 105.3? |
||
Response 4: Thank you very much for your comment. We are sorry for the mistake, and have revised the ‘105.3’ to ‘105.3’ in revised manuscript (Line 184 in revised manuscript, previously Line 166). |
||
Comments 5: In your phylogenetic diagrams in Fig 3, did not see your isolate BJ0304 in the list? |
||
Response 5: Thank you very much for your comment. We are sorry for make you confused. The isolate BJ0304 was highlighted with a green solid circle in Figure 3 (OM363683 BJ0304 China 2021 pig). |
||
Comments 6: Is Getah virus normally pathogenic in mice or does it mostly result in asymptomatic infections? Your speculation that the E3 and nsp3 may be involved in pathogenicity is interesting and you do provide some preliminary evidence for this. However, there may be other reasons for why your isolate is not pathogenic in mice, can you suggest any other reasons? |
||
Response 6: We appreciate the comments. According to previous publications, GETV can cause different scenarios in mice varying from asymptom to lethal disease, which depends on the viral virulence. The contamination may have been present for a long time, which caused GETV to be attenuated during the successive passage of vaccine strain. Meanwhile, by comparing differences among GETV strains in this study, we found specific mutation/glycosylation of nsP1/P2/P3/P4, C and E2, and speculated the possible relationship between E2/nsp3 and viral virulence based on what is known about the role of protein from alphaviruses. However, from perspective of mutation/glycosylation alone, nsP1/P2/P3/P4 may also be associated with the viral virulence by exerting some certain immune evasion function. In addition to the amino acid level, the difference of nucleic acid level (e.g. lncRNA, circRNA) and protein post-translational modification level may also be the virulent factor. |
||
Comments 7: Finally in your title, should it not be ".... from a commercial modified-live virus vaccine." |
||
Response 7: Thank you very much for your comment. According to your advice, the title has been revised to ‘Isolation and characterization of a novel group III-classified Getah virus from a commercial modified-live vaccine against PRRSV’ in revised manuscript (Line 2-4 in revised manuscript, previously Line 2-3). |
||
4. Response to Comments on the Quality of English Language |
||
Point 1: Very good overall, some minor editing would improve the communication in certain areas. |
||
Response 1: We appreciate the suggestion and performed the correction and rewriting of certain words and sentences to improve readability of the manuscript. |